# Prospective Comparison of Nine Different Handheld Ultrasound (HHUS) Devices by Ultrasound Experts with Regard to B-Scan Quality, Device Handling and Software in Abdominal Sonography

**DOI:** 10.3390/diagnostics14171913

**Published:** 2024-08-30

**Authors:** Daniel Merkel, Christian Lueders, Christoph Schneider, Masuod Yousefzada, Johannes Ruppert, Andreas Weimer, Moritz Herzog, Liv Annebritt Lorenz, Thomas Vieth, Holger Buggenhagen, Julia Weinmann-Menke, Johannes Matthias Weimer

**Affiliations:** 1BIKUS—Brandenburg Institute for Clinical Ultrasound, Brandenburg Medical School Theodor Fontane (MHB), 16816 Neuruppin, Germany; daniel.merkel@mhb-fontane.de; 2Immanuel Klinik Rüdersdorf, University Hospital of the Brandenburg Medical School, 15562 Rüdersdorf bei Berlin, Germany; 3Klinik am See, Rehabilitation Center for Internal Medicine, 15562 Rüdersdorf bei Berlin, Germany; 4Department of Medicine, Justus Liebig University Giessen, 35385 Giessen, Germany; 5Center of Orthopedics, Trauma Surgery, and Spinal Cord Injury, Heidelberg University Hospital Heidelberg, 69118 Heidelberg, Germany; 6Else Kröner Fresenius Center for Digital Health, Faculty of Medicine Carl Gustav Carus, Technical University Dresden, 01307 Dresden, Germany; 7Department of Radiooncology and Radiotherapy, Mainz University Hospital, 55131 Mainz, Germany; 8Rudolf Frey Learning Clinic, University Medical Center of the Johannes Gutenberg, University Mainz, 55131 Mainz, Germany; 9I. Department of Medicine, University Medical Center of the Johannes Gutenberg, University Mainz, 55131 Mainz, Germany

**Keywords:** pocket device, handheld ultrasound, POCUS, comparison, handling, software, B-mode quality

## Abstract

Background: The HHUS market is very complex due to a multitude of equipment variants and several different device manufacturers. Only a few studies have compared different HHUS devices under clinical conditions. We conducted a comprehensive prospective observer study with a direct comparison of nine different HHUS devices in terms of B-scan quality, device handling, and software features under abdominal imaging conditions. Methods: Nine different HHUS devices (Butterfly iQ+, Clarius C3HD3, D5CL Microvue, Philips Lumify, SonoEye Chison, SonoSite iViz, Mindray TE Air, GE Vscan Air, and Youkey Q7) were used in a prospective setting by a total of 12 experienced examiners on the same subjects in each case and then assessed using a detailed questionnaire regarding B-scan quality, handling, and usability of the software. The evaluation was carried out using a point scale (5 points: very good; 1 point: insufficient). Results: In the overall evaluation, Vscan Air and SonoEye Chison achieved the best ratings. They achieved nominal ratings between “good” (4 points) and “very good” (5 points). Both devices differed significantly (*p* < 0.01) from the other seven devices tested. Among the HHUS devices, Clarius C3HD3 and Vscan Air achieved the best results for B-mode quality, D5CL Microvue achieved the best results for device handling, and SonoEye Chison and Vscan Air achieved the best results for software. Conclusions: This is the first comprehensive study to directly compare different HHUS devices in a head-to-head manner. While the majority of the tested devices demonstrated satisfactory performance, notable discrepancies were observed between them. In particular, the B-scan quality exhibited considerable variation, which may have implications for the clinical application of HHUS. The findings of this study can assist in the selection of an appropriate HHUS device for specific applications, considering the clinical objectives and acknowledging the inherent limitations.

## 1. Introduction

Recently, ultrasound (US) has gained importance in clinical routine and preclinical emergency diagnostics. The minimization and improved digitization of equipment have led to the development of very compact ultrasound devices in the past two decades, which have been established as handheld ultrasound (HHUS) devices in a wide range of medical applications. In particular, from the perspective of point-of-care diagnostics (POCUS) [1], HHUS devices play an important role not only in the field of emergency medicine [2,3], outpatient diagnostics [4,5] and bedside diagnostics [6,7,8] but also under isolation conditions [9], in intensive care medicine [10] and, last but not least, in sonography training [11,12]. Several professional societies have published guidelines on the indication and safety of using HHUS [13,14,15,16].

While there are a few established manufacturers of traditional US devices [17], the variety of available HHUS systems [18] is increasing in parallel with their increasing economic importance [19,20,21]. There are both wired and wireless systems, HHUS with universal “all in one” transducers, as well as HHUS with several attachable transducers and systems that combine two commonly used transducer variants (“two-in-one” devices) [18,22]. The variety of available HHUS devices with very specific functionalities makes the selection of a suitable HHUS for the specific medical application difficult.

Several aspects come into play when selecting a suitable HHUS device [22,23], of which the specific medical question and the environment of the HHUS are the most important. For example, in a pre-clinical emergency setting, the demands placed on an HHUS device are different from those in an infectious disease intensive care unit under isolated conditions or in rural or underserved areas for general medical purposes [4,5,24].

In addition to equipment, weight, transducer shape, battery life, start-up time, heat generation, and other parameters, sufficient B-image quality is undoubtedly a primary criterion when assessing different HHUS devices [15,25]. Very few studies have compared HHUS devices from different manufacturers in terms of inter-system variability [26,27,28,29]; only two have performed an actual head-to-head comparison of HHUS devices on the same test subjects [28,29]. In a gynecological pilot study, three different HHUS devices were compared by the same examiner [29] regarding measurement accuracy. In a further study, four different HHUS devices used by several experts on the same test subject exhibited no significant differences regarding handling, but one of the devices was inferior in terms of B-image quality [28].

This prospective study evaluates a large variety of HHUS devices in a comprehensive head-to-head comparison by several experts. The primary goal is to make a direct comparison of B-image quality, handling, and user-friendliness of the software for a total of nine different HHUS devices in clinical abdominal US. The secondary goal is to rank the devices by cumulatively considering the various evaluation aspects.

## 2. Material and Methods

### 2.1. Study Design

This prospective observational study comparing different HHUS devices with regard to B-image quality, handling, and user-friendliness was conducted in July 2024 in a German university hospital. Ultrasound device manufacturers supplied their HHUS devices to a clinical ultrasound laboratory for a period of one week. Ultrasound experts were invited to test the devices on healthy subjects and to evaluate them using a structured questionnaire. The study protocol was reviewed and accepted by the Ethics Committee of the Brandenburg Medical School under the reference number E-01-20220502.

### 2.2. HHUS Devices

HHUS devices were defined as ultra-compact portable sonography devices consisting only of a transducer, with no peripheral devices, that can be operated with one hand [17,24]. The sonography images generated are transmitted to a separate display either by wire or wirelessly. These devices are approved as medical devices and have a CE label (label for European conformity in commercial products). The device had to be available for a trial period within a previously clearly defined period (1 week). In addition to an internet and literature search [18,26,27,28,30], several sonography device dealers and three university ultrasound centers were contacted to scope the HHUS market. Nine of the total 14 HHUS devices identified this way were included in this study (Table 1; Figure 1 and Figure 2). For three devices, no dealer contact could be established (iSiniQ 30A, mSonics MU1, and Sonostar Uprobe-C4PL), and the remaining two manufacturers (Alpinion minisono and Kosmos) could not supply their devices for the defined study period.

### 2.3. Examiners

The work of Howard et al. was used to plan the number of raters. A minimum number of 10 raters was defined [31]. The examiners had to be certified medical specialists, complete the testing of all available devices, and consent to participate in this study. They were recruited from an existing collegial network of DEGUM members (German Society for Ultrasound in Medicine).

### 2.4. Testing of Devices

Several testing supervisors were extensively trained on all devices. All testing took place in the same clinical ultrasound laboratory. All nine HHUS devices included in this study were displayed next to each other in open boxes. Each device had its own display (iPad or tablet) with the current device software installed. The devices were at full electrical charge. Testing was performed in a random order. Before each test, a supervisor switched on the HHUS, connected it to its display, and provided basic operating instructions. During the examinations, the displays were on a separate stand to the left of the test subjects, as is customary in ultrasound examinations. The supervisor was also available during all evaluations later on and assisted with the device settings. The examiner was asked to test each HHUS extensively considering all test parameters on a healthy test subject without a specified time limit and to fill out an evaluation form immediately afterwards. Finally, the examiners were asked to name the top three devices that were their personal favorites based on the overall impression of all criteria.

### 2.5. Evaluation Form

An interdisciplinary team (ultrasound experts and didactics) designed a written questionnaire based on the current literature containing three main items, “image quality”, “handling”, and “software”, with multiple graded subitems. The assessment was based on a 5-point Likert scale (5 = very good; 4 = good; 3 = satisfactory; 2 = sufficient; 1 = unsatisfactory).

### 2.6. Statistical Analysis

The data were entered by transferring the evaluation forms into an Excel spreadsheet (Microsoft Excel^®^ Version 16.48, Microsoft Corporation, Redmond, WA, USA). All statistical analyses were performed in Rstudio (Rstudio Team [2020]. Rstudio: Integrated Development for R. Rstudio, PBC, http://www.rstudio.com, last accessed on 15 July 2024) with R 4.0.3 (A Language and Environment for Statistical Computing, R Foundation for Statistical Computing, http://www.R-project.org; last accessed on 15 July 2024). Binary and categorical baseline variables are given as absolute numbers and percentages. Continuous data are given as median and interquartile range (IQR) or as mean and standard deviation (SD). The D’Agostino and Pearson test was used to test for normal distribution. Categorical variables were compared using the chi-squared test, and continuous variables were compared using the T-test or the Mann–Whitney U test. Parametric (ANOVA) or non-parametric (Kruskal–Wallis) analyses of variance were calculated and further explored with pairwise post hoc tests (*t*-test or Mann–Whitney U). *p*-values of <0.05 were considered statistically significant. We also used the intraclass correlation coefficient (ICC) type 3,k with the values ranging from 0 (no reliability) to 1 (perfect reliability) to check the consistency between measures of the same class. We have also calculated Krippendorff’s alpha from the raters and the evaluation criteria.

## 3. Results

### 3.1. Examiners

A total of 12 examiners were recruited for this study. Appendix A shows their baseline characteristics. Most (67%) work in internal medicine and have more than 10 years of professional experience (75%). All had already performed at least 1000 independent ultrasound examinations and are currently performing more than 20 examinations per week in diagnostic ultrasound, in either out-patient (42%) or in-patient settings (58%). Most had no previous experience (75%) with HHUS.

### 3.2. Results of the Ratings

Table 2 and Figure 3 show the results of the ratings for the main items and subitems. The average grade of the main items and subitems was 3.8, with a 95 percent confidence interval (95% CI) between 3.6 and 3.8. This corresponds to a nominal assessment of all evaluation items between “good” (4 points) and “satisfactory” (3 points). Significant differences were identified between the individual devices and categories, which are explained in more detail below for each category and device.

### 3.3. B-Mode Image Quality

Table 2 and Figure 3a show the grading for the subitems of B-mode image quality. The average grade was 3.7, with a 95 percent confidence interval (95% CI) between 3.5 and 3.8. This corresponds to a nominal assessment of all evaluation items between “good” (4 points) and “satisfactory” (3 points).

The best results in B-mode quality were achieved using the Clarius C3HD3 (4.3, 95% CI 4.0 to 4.5), Vscan Air (4.2, 95%CI 4.0 to 4.4), and Chison SonoEye (3.9, 95% CI 3.6 to 4.2) devices. This corresponds to a nominal assessment of all evaluation items between “very good” (5 points) and “good” (4 points). The other devices were graded significantly worse than these top three. All significance levels are shown in Appendix A.

### 3.4. Handling

Table 2 and Figure 3b show the grading for device handling. The average grade was 3.7, with a 95 percent confidence interval (95% CI) between 3.6 and 3.9. This corresponds to a nominal assessment of all evaluation items between “good” (4 points) and “satisfactory” (3 points).

The best results in device handling were achieved by D5CL Microvue (4.4, 95% CI 4.2 to 4.5). This corresponds to a nominal assessment of all evaluation items between “very good” (5 points) and “good” (4 points). This grade is significantly better than that of the other devices. All significance levels are shown in Appendix A.

### 3.5. Software

Table 2 and Figure 3c show the grading for the software. The average rating was 3.7, with a 95 percent confidence interval (95% CI) between 3.5 and 3.8. This corresponds to a nominal assessment of all evaluation items between “good” (4 points) and “satisfactory” (3 points).

The best grades for the software were achieved by the SonoEye Chison (4.4, 95% CI 4.3 to 4.5) and Vscan Air (4.2, 95% CI 4.1 to 4.4) devices. This corresponds to a nominal assessment of all evaluation items between “very good” (5 points) and “good” (4 points). Both devices received significantly higher grades than the others. All significance levels are shown in Appendix A.

### 3.6. Overall Value

Table 2 and Figure 3d show the cumulative grades for all subitems.

The best overall grades were achieved by the Vscan Air (4.1, 95% CI 4.0 to 4.2) and Sonoeye Chison (4.1, 95% CI 3.9 to 4.2) devices. This corresponds to a nominal assessment of all evaluation items between “very good” (5 points) and “good” (4 points). Both devices received significantly higher grades than the others. All significance levels are shown in Appendix A.

### 3.7. Final Subjective Overall Assessment

Figure 4 shows the results of the evaluators’ final subjective selection of the three best devices. Half of the evaluators (50%) voted the Vscan Air in first place. Further, two-thirds of the evaluators voted this device in the top three ratings. The Clarius C3HD3 and Chison SonoEye devices also achieved good ratings. These devices were selected as the top three in half of all evaluations.

### 3.8. Inter-Category Comparison per Device

An inter-category comparison per device is shown in Figure 5 and Figure 6.

The grades achieved in the three main categories, B-mode quality, handling, and software, were significantly correlated, particularly for the Philips Lumify, Vscan Air, and Youkey devices. Notable discrepancies were evident between the D5CL Microvue and TE Air devices. In the case of both devices, the handling was rated as very good, whereas the B-scan quality was only average.

Vscan Air was the only device for which the three main parameters correlated with each other and received high grades. This was also reflected in the final overall assessment of the devices made by the examiners, in which half of all examiners (50%) selected the Vscan Air as their favorite in a final subjective overall assessment (Figure 4).

A presentation of the main parameters as a function of the tested device can be found in Figure 5.

The ICC values for most devices are relatively high (ranges between 0.77–0.95), indicating good to excellent rater reliability (see Appendix A Grading of HHUS devices overall, per main item overall and per subitem shown as median and IQR.). Krippendorff’s alpha for the 12 raters and 9 subjects was 0.23. The exclusion of extreme ratings shows an increase up to a Krippendorff’s alpha of 0.31 for 10 raters and 0.47 for 8 raters.

## 4. Discussion

The quality of nine different HHUS devices was evaluated comprehensively by ultrasound experts for the first time in this study. It was found that the overall B-mode quality of HHUS is better than previously estimated [6,32,33,34,35,36]. It is apparent that there are significant differences in B-scan quality, device handling, and software between HHUS devices from different manufacturers.

The use of HHUS systems for specific questions, as well as in comparison with conventional US systems [30,37,38], has been evaluated more than their inter-system variability. Preliminary studies on HHUS for intensive care and internal medicine detected potentially clinically relevant differences in the B-image quality of different HHUS devices. However, the assessments were not based on a direct head-to-head comparison of HHUS devices but rather on parallel examinations of a high-end device [26,27]. To date, comparisons have been made regarding a maximum of four different HHUS devices on the same test subjects. These were limited to measurement accuracy [29], ease of use [28], and overall satisfaction [28]. We included nine HHUS devices and many more parameters in this head-to-head comparison.

### 4.1. B-Image Quality

B-image quality, and, thus, the clinical significance of a sonographic examination, depends on many variables, for example, not only on the expertise of the examiner [39,40], the test subject’s ultrasound conditions, and the quality of the device or transducer [41] but also on the presets and post-processing [42,43,44]. Although measurements of phantoms have been established for standardization [45,46] and are required by professional societies [47], assessments of sonographic B-image quality in clinical examinations are predominantly carried out subjectively [23,26]. A systematic study of various HHUS devices showed that their B-image quality can match that of cart-based systems based on phantom measurements but does not correspond to that of high-end devices in terms of perceived detail resolution [48]. The present study did not consider the use of US phantoms to assess B-scan quality, as the primary objective was to investigate the clinical applications of HHUS. Future comparative studies should include systematic standardized US phantoms.

Cart-based US systems use various transducers shaped for their intended use. A convex transducer produces better B-images for abdominal sonographic questions than an “all-in-one” transducer aimed at a plethora of clinical settings. Two of the nine devices achieved the best B-image quality. Both are equipped with separate convex and linear transducers (Clarius C3HD3 and Vscan Air). Both HHUS devices have already impressed in previous studies with regard to B-image quality in abdominal US [26,27,28,48]. Both HHUS devices tested with a universal transducer (Butterfly iQ+ and TE Air) were inferior in terms of B-image quality. A special feature of Butterfly iQ+ is worth mentioning, as it is the only HHUS that uses a special microchip technology as a transducer instead of piezo crystals [1,22].

### 4.2. Handling and Software

The combination of a transducer, battery, and electronics in one device leads to various shapes, sizes, and weights of these ultra-compact “pocket-sized” HHUS devices (Figure 1).

In a previous study, there were no significant differences in handling between the four devices tested [28]. In contrast, the handling of the Microvue D5CL HHUS was rated significantly better than that of the other eight devices in this study. It is a wireless, extremely compact, lightweight device with two opposing ultrasound transducers. Interestingly, the presence of a wired connection to the monitor had no influence on the rating of handling. The three wired HHUS devices (Philips Lumify, Chison SonoEye, and Butterfly iQ+) ranked third, fifth, and sixth, with no significance to the other wireless devices.

All devices can connect to the internet, allowing for an upgrade or update of the software [41]. Nevertheless, HHUS devices have to meet particularly high requirements in terms of the user-friendliness of the software, as, unlike cart-based US devices, they do not have a conventional control panel for continuous image optimization [42,43,44]. For image optimization, the settings need to be adjusted via a menu within the software, thus requiring good preset settings, which were evaluated separately in this study. Usually, the free hand not holding the transducer is used for image optimization at the cart. In HHUS, this free hand is often occupied with holding the monitor [25], impeding manual image optimization. Solutions to this problem such as Velcro fasteners or tablet stands for the monitors often contradict the “pocket size” feature of HHUS.

SonoEye and the Vscan Air had the best software according to the evaluators. Even though the majority of the software gradings in this study were “acceptable” or “good”, the software was graded “inadequate” by individual reviewers for five of the nine devices tested. One argument to be made here is that, with longer and more frequent use of a specific HHUS device, one can become familiarized with less intuitive software and find hidden submenus more quickly. There was no time limit for performing the tests in this study. Despite the absence of a planned time recording procedure in accordance with the study protocol, each HHUS was tested for an approximate period of between five and ten minutes. This duration is sufficient for the user to become acquainted with the device. However, it is unlikely to be sufficient for an exhaustive exploration of the software’s submenus. We are not aware of any previous studies evaluating HHUS software.

### 4.3. Overall Satisfaction

As a final part of the assessment, each examiner had to name their top three of the nine HHUS devices tested (Figure 4). We found that 50% of the evaluators chose Vscan Air in first place and in the top three in two-thirds of the cases. This is similar to previous head-to-head comparisons [28] and is consistent with the results of comparative studies with a high-end device [26,27]. The runner-ups were Clarius C3HD3 and Chison SonoEye, each with 50% of the examiners placing them within the top three ratings. The spontaneous final ranking of the devices correlated very well with the numerically determined overall grading (Figure 3).

### 4.4. Clinical Importance and Perspective

The omnipresent availability of HHUS to medical professionals, as well as increasingly to non-medical professionals, raises questions regarding the limitations and risks of its use. Several position papers from professional societies emphasize the adequate training and expertise of users as prerequisites for the clinical use of HHUS [13,15,49].

It has been proven that assessments such as those of ascites or pleural effusion in an outpatient palliative setting or after cardiac surgery can certainly be carried out with HHUS by non-physician staff [5]. However, one cannot expect to be able to perform difficult oncological diagnostics with HHUS, e.g., in the context of hepatocellular cancer (HCC) screening in cirrhosis patients [50,51]. However, as the B-scan quality of HHUS increases, its field of application will expand to areas that have so far been reserved for high-end sonography. A sharp boundary between the areas of application of HHUS and conventional high-end US has not yet been defined [41]. However, HHUS certainly has much more potential than just detecting liquids [2,3,34]. The use of ultrasound in orthopedic and respiratory clinical settings could be increasingly implemented using HHUS in the future [52,53].

Due to their good networking and continuous possibility of cloud-based data storage, HHUS devices offer ideal conditions for collecting large amounts of data and, therefore, artificial intelligence (AI)-based applications. Several potential applications have already been published, mostly in the form of pilot studies [54,55]. Further developments can be expected in this area in the foreseeable future, which may fundamentally revolutionize the entire field of imaging diagnostics.

### 4.5. Use in Ultrasound Training

The positive evaluation of HHUS devices in the present study could also be used as an opportunity to adapt future course concepts in ultrasound training.

More integration of HHUS into short formats would enable even greater location-independent learning and teaching [56,57]. In undergraduate sonography training, which has increased significantly in recent years [58,59,60,61,62], HHUS devices could serve as a useful supplement and alternative to conventional ultrasound devices and, thus, reduce the expenditure of resources due to lower acquisition costs, allowing for more widespread implementation [18,63]. Their use in low- and middle-income countries in point-of-care diagnostics may lead to the implementation or maintenance of basic diagnostic capabilities in poorer regions [64,65].

## 5. Limitations

Not all available devices could be included. Even though the examiners were very experienced, the sample of examiners was very small. Other factors, such as pricing, durability, heat development during prolonged use, and more, were not evaluated. Some examiners had prior experience with HHUS. The device manufacturers were visible on the devices or in the software interface. Each examiner performed all testing on the same test subject for all devices, but not all examiners had the same test subject. Although the devices were each used on the same test subject, it was not mandatory to perform and compare the identical sonographic sections with each device. Another limitation of this study is the discrepancy between the ICC and Krippendorff’s alpha values in the inclusion of 12 raters, due to different assumptions and calculation bases. These differences make a consistent interpretation of inter-rater reliability difficult. To resolve this discrepancy in future studies, an adapted methodology could be used that integrates and harmonizes both measures, e.g., by using adapted weighting strategies or a more differentiated analysis of rater agreement.

## 6. Conclusions

Handheld ultrasound devices are becoming increasingly prevalent in clinical practice. They are useful when time is a critical factor (in emergency rooms and intensive care units) or when the setting is conducive to portable devices (remote locations, isolation ward, ambulatory use, isolation ward, etc.). This is the first comprehensive study to directly compare nine different HHUS devices in a head-to-head manner. While the majority of the tested devices demonstrated sufficient performance, there were notable discrepancies between them. In particular, the B-scan quality varied considerably, which has implications for the clinical application of HHUS. The findings of this study will assist in the selection of an appropriate HHUS device for specific applications, considering the clinical objectives and acknowledging the inherent limitations.

## Figures and Tables

**Figure 1 diagnostics-14-01913-f001:**
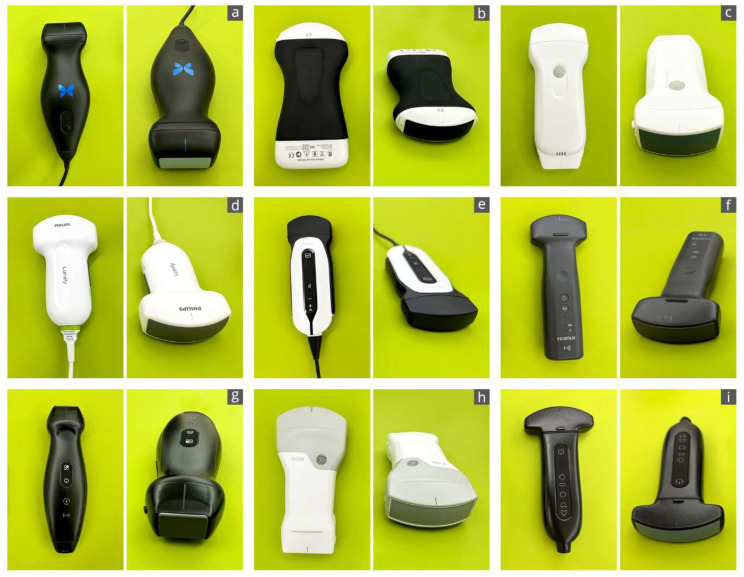
Overview of the devices included in this study: (**a**): Butterfly iQ+; (**b**): Clarius C3HD3; (**c**): D5CL Microvue; (**d**): Philips Lumify; (**e**): SonoEye; (**f**): SonoSite iViz; (**g**): TE Air; (**h**): Vscan Air; (**i**): Youkey Q7.

**Figure 2 diagnostics-14-01913-f002:**
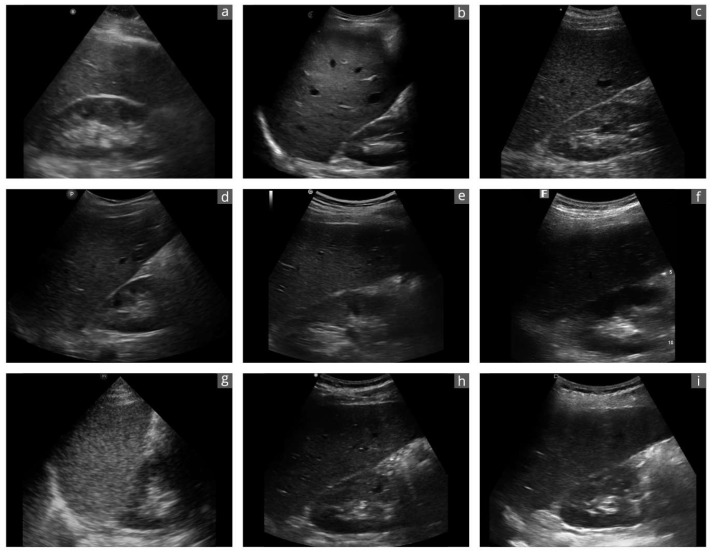
Exemplary sonographic images of a liver–kidney section of the same subject with the different devices: (**a**): Butterfly iQ+; (**b**): Clarius C3HD3; (**c**): D5CL Microvue; (**d**): Philips Lumify; (**e**): SonoEye; (**f**): SonoSite iViz; (**g**): TE Air; (**h**): Vscan Air; (**i**): Youkey Q7. Due to the transducer ergonomics and device settings, the images are not exactly identical.

**Figure 3 diagnostics-14-01913-f003:**
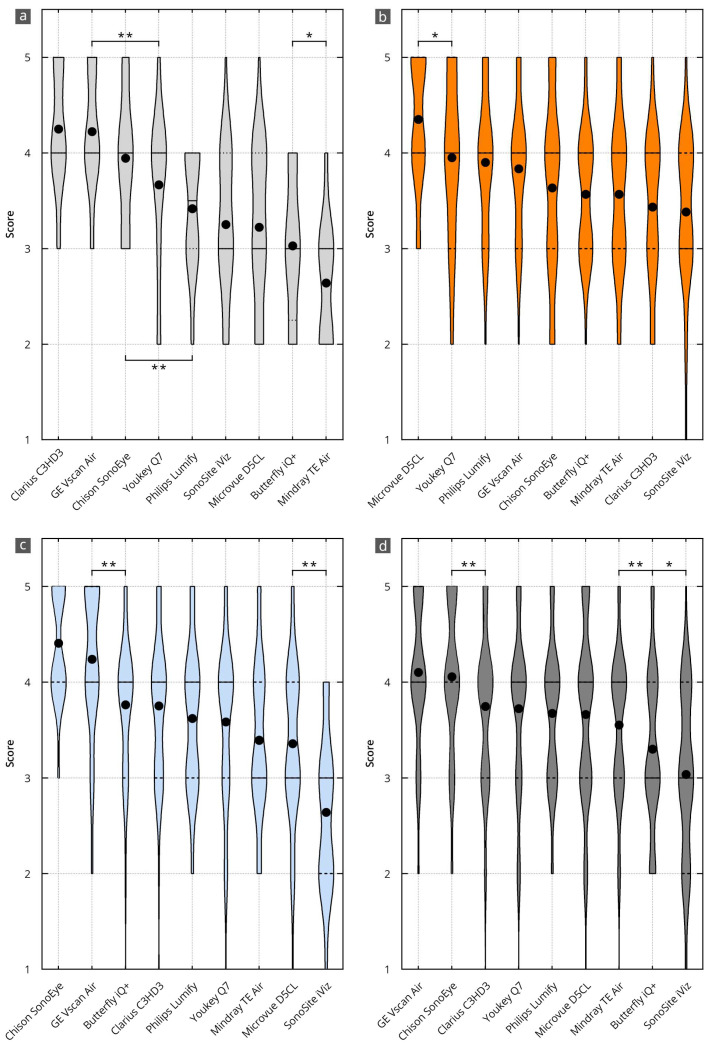
Results of the evaluation of the HHUS devices in relation to the items surveyed in the main categories of B-mode image quality (**a**), device handling (**b**), software (**c**), and overall grade (**d**). The order in which the individual HHUS devices are displayed was determined by the grade achieved. The black dot indicates the mean value. X^(* = *p* < 0.05; ** = *p* < 0.01). A complete list of the *p*-values can be found in Appendix A.

**Figure 4 diagnostics-14-01913-f004:**
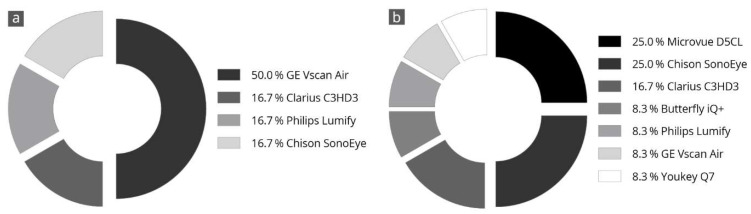
Specification of a personal favorite after completion of all tests; the devices most frequently in 1st place (**a**) and 2nd place (**b**) are listed. The totals may not add up exactly to 100% due to rounding.

**Figure 5 diagnostics-14-01913-f005:**
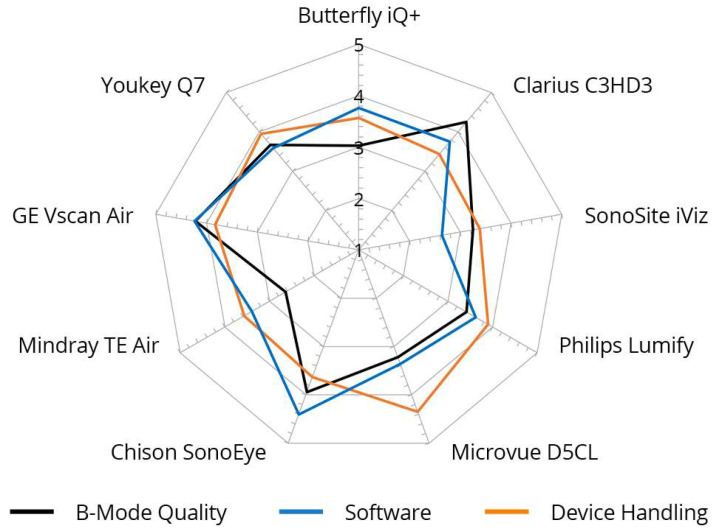
Inter-category comparison per device with regard to B-mode quality, device handling, and software, cumulative for nine different devices. Graded from 1 (unsatisfactory) to 5 (very good).

**Figure 6 diagnostics-14-01913-f006:**
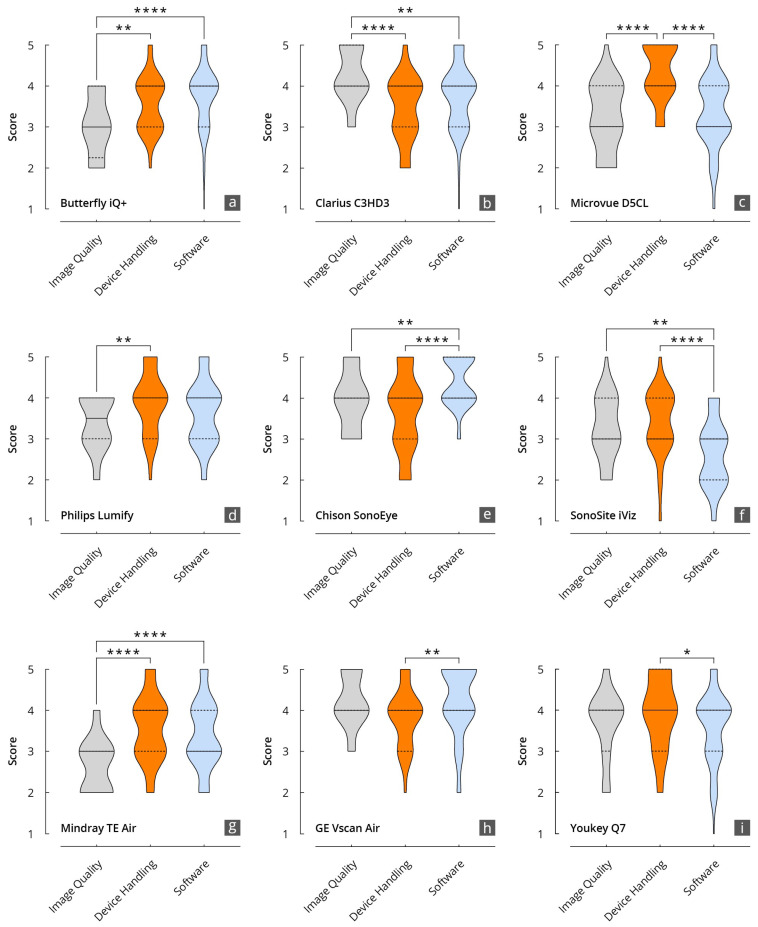
Results of the evaluation of the HHUS devices in relation to the items surveyed in the main categories of B-mode image quality, device handling, and software. (**a**): Butterfly iQ+; (**b**): Clarius C3HD3; (**c**): D5CL Microvue; (**d**): Philips Lumify; (**e**): SonoEye; (**f**): SonoSite iViz; (**g**): TE Air; (**h**): Vscan Air; (**i**): Youkey Q7 (* *p* < 0.05; ** *p* < 0.01; **** *p* < 0.0001).

**Table 1 diagnostics-14-01913-t001:** HHUS devices requested for the study (in alphabetical order).

	Device Name	Manufacturer	City/Country	Image Transmission	Transducer
1	Alpinion minisono *	Alpinion Medical Systems	Seoul, Republic of Korea	Wired	Linear or convex
2	Butterfly iQ+	Butterfly Network, Inc.	Burlington, MA, USA	Wired	“All-in-one 1.75 D-Array”
3	Clarius C3HD3	Clarius Mobile Health Corp.	Burnaby, BC, Canada	Cordless	Linear or convex
4	D5CL Microvue	Guangzhou SonoHealth Medical Technologies Co., Ltd.	Guangzhou, China	Cordless	Linear and convex in one
5	iSiniQ 30A *	Prunus Medical Shenzhen	Shenzhen, China	Cordless	Two attachable probes
6	Kosmos *	EchoNous EchoNous Inc.	Redmond, WA, USA	Wired	All-in-one transducer
7	mSonics MU 1 *	Lang Sheng Sozhou	Guangdong, China	Wired	Convex
8	Philips Lumify	Koninklijke Philips N.V.	Amsterdam, The Netherlands	Wired	Linear, convex, or broadband convex
9	SonoEye	CHISON Medical Technologies Co., Ltd.	Wuxi, China	Wired	Linear, convex, sector, or broadband convex
10	SonoSite iViz	FUJIFILM Corporation	Tokyo, Japan	Cordless	Linear or convex
11	Sonostar Uprobe-C4PL *	Universal Diagnostic Solutions	Vista, CA, USA	Cordless	Linear and convex in one
12	TE Air	Shenzhen Mindray Bio-Medical Electronics Co., Ltd.	Shenzhen, China	Cordless	Sector
13	Vscan Air	General Electric	Boston, MA, USA	Cordless	Convex/linear or sector/linear in one
14	Youkey Q7	Wuhan Youkey Bio-Medical Electronics Co., Ltd.	Wuhan, China	Cordless	Four different replaceable transducers

The devices marked with an asterisk (*) were not included in the study due to a lack of response from the manufacturer or because no devices were available at the time of the study.

**Table 2 diagnostics-14-01913-t002:** Grading of HHUS devices overall, per main item overall, and per subitem shown as average grade (±standard deviation). An additional presentation of the results as median and IQR can be found in Appendix A.

Item		Butterfly iQ+	Clarius C3HD3	Sono-Site iViz	Philips Lumify	D5CL Microvue	SonoEye Chison	TE Air Mind-ray	Vscan Air GE	Youkey Q7	*p*-Value
All items		3.6 ± 0.8	3.7 ± 0.8	3.0 ± 0.9	3.7 ± 0.8	3.7 ± 0.9	4.1 ± 0.8	3.3 ± 0.9	4.1 ± 0.8	3.7 ± 0.9	<0.0001
B-mode image quality	**Overall**	3.0 ± 0.7	4.3 ± 0.7	3.3 ± 0.8	3.4 ± 0.7	3.2 ± 0.8	3.9 ± 0.7	2.6 ± 0.6	4.2 ± 0.6	3.7 ± 0.8	<0.0001
	Resolution	3.0 ± 0.9	4.3 ± 0.7	3.1 ± 0.8	3.5 ± 0.7	3.2 ± 0.9	3.9 ± 0.8	2.5 ± 0.5	4.3 ± 0.5	3.7 ± 0.9	<0.0001
	Contrast	3.0 ± 0.7	4.2 ± 0.7	3.3 ± 0.6	3.3 ± 0.6	3.3 ± 0.9	4.0 ± 0.7	2.7 ± 0.7	4.0 ± 0.7	3.5 ± 0.9	<0.0001
	Overall impression	3.1 ± 0.7	4.3 ± 0.6	3.4 ± 0.9	3.5 ± 0.7	3.3 ± 0.8	3.9 ± 0.7	2.8 ± 0.8	4.3 ± 0.7	3.8 ± 0.7	<0.0001
Handling	**Overall**	3.6 ± 0.7	3.4 ± 0.8	3.4 ± 0.8	3.9 ± 0.8	4.4 ± 0.7	3.6 ± 0.9	3.6 ± 0.8	3.8 ± 0.7	4.0 ± 0.9	<0.0001
	Haptics	3.5 ± 0.5	3.3 ± 0.7	3.4 ± 0.7	4.2 ± 0.7	4.6 ± 0.5	3.2 ± 0.7	3.6 ± 0.8	4.0 ± 0.6	3.9 ± 0.8	<0.0001
	Weight	3.2 ± 0.8	2.9 ± 0.9	3.8 ± 0.7	4.3 ± 0.6	4.6 ± 0.5	4.2 ± 0.8	3.8 ± 0.9	3.8 ± 0.8	4.2 ± 0.7	<0.0001
	Shape	3.7 ± 0.7	3.3 ± 0.7	3.1 ± 0.9	3.9 ± 0.7	4.6 ± 0.5	3.1 ± 0.8	3.5 ± 0.9	3.8 ± 0.8	3.9 ± 1.0	0.0005
	Connectivity	4.0 ± 0.6	3.9 ± 0.7	3.5 ± 0.5	3.3 ± 0.9	4.0 ± 0.7	4.0 ± 1.0	3.6 ± 0.7	3.7 ± 1.0	3.9 ± 0.8	0.23
	Overall impression	3.5 ± 0.5	3.7 ± 0.8	3.1 ± 1.0	3.8 ± 0.6	4.0 ± 0.9	3.8 ± 1.0	3.4 ± 0.9	4.0 ± 0.4	3.8 ± 1.1	0.21
Software	**Overall**	3.8 ± 0.7	3.7 ± 0.8	2.6 ± 0.8	3.6 ± 0.8	3.4 ± 0.9	4.4 ± 0.6	3.4 ± 0.8	4.2 ± 0.8	3.6 ± 0.9	<0.0001
	Presets	3.8 ± 0.7	3.9 ± 0.5	2.3 ± 0.8	3.6 ± 0.8	3.4 ± 0.7	4.5 ± 0.5	3.4 ± 0.9	4.0 ± 0.9	3.8 ± 0.7	<0.0001
	Depth	3.9 ± 0.8	3.9 ± 0.7	2.7 ± 0.8	3.7 ± 0.7	2.7 ± 1.0	4.5 ± 0.5	3.5 ± 1.0	4.3 ± 1.1	3.6 ± 0.9	<0.0001
	Gain	4.1 ± 0.5	3.9 ± 0.7	2.8 ± 0.7	3.7 ± 0.9	3.4 ± 0.8	4.3 ± 0.7	3.7 ± 1.0	4.4 ± 0.9	3.4 ± 1.1	0.0002
	Duplex/PRF	3.6 ± 1.1	3.3 ± 0.7	-	3.3 ± 0.6	3.4 ± 1.0	4.3 ± 0.7	3.2 ± 0.8	4.3 ± 0.6	3.5 ± 0.8	0.0008
	Saving	3.8 ± 0.6	3.7 ± 1.2	3.1 ± 0.9	3.8 ± 0.6	3.8 ± 0.7	4.4 ± 0.8	3.4 ± 0.9	4.0 ± 0.7	3.6 ± 0.9	0.04
	Intuitiveness	3.5 ± 0.7	3.8 ± 0.9	2.7 ± 0.7	3.7 ± 1.0	3.3 ± 0.9	4.3 ± 0.5	3.4 ± 0.7	4.5 ± 0.7	3.6 ± 0.8	<0.0001
	Overall impression	3.6 ± 0.7	3.8 ± 0.8	2.2 ± 0.9	3.6 ± 0.9	3.4 ± 0.7	4.4 ± 0.5	3.2 ± 0.6	4.2 ± 0.9	3.6 ± 0.9	<0.0001

## Data Availability

The data presented in this study are available on request from the corresponding author. The data are not publicly available because of institutional and national data policy restrictions imposed by the ethics committee since the data contain information that could potentially identify study participants. Data are available upon request (contact via weimer@uni-mainz.de) for researchers who meet the criteria for access to confidential data (please provide the manuscript title with your inquiry).

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
