# Peer review of "Prospective Comparison of Nine Different Handheld Ultrasound (HHUS) Devices by Ultrasound Experts with Regard to B-Scan Quality, Device Handling and Software in Abdominal Sonography"

_diagnostics, 2024, doi:10.3390/diagnostics14171913_

Round 1

Reviewer 1 Report

Comments and Suggestions for Authors

The authors evaluated nine ultra-compact handheld ultrasound (HHUS) devices in a prospective setting by a total of 12 experienced radiologists. This article is useful for the clinical practice. However, some modification is required for the final publication.

Its better to evaluate the image quality using the standard ultrasound imaging phantom for reasonable comparison.

Although sonographic images of a liver-kidney section of the same subject with the different devices are shown in Figure 2, the position is not exactly the same for easy comparison.

Several popular abdominal examination locations are preferred for comparison in order to demonstrate the general purpose performance.

Change there are few established manufacturers to there are a few established manufacturers 

Are data in Figure 5 same as those in Figure 3?

Comments on the Quality of English Language

overal it is good

Author Response

Dear reviewer, thank you very much for reviewing our manuscript. Please find attached a P2P of your comments, which we hope we have addressed and implemented to your satisfaction. 

Reviewer 2 Report

Comments and Suggestions for Authors

This is an interesting prospective study comparing 9 HHUS in healthy volunteers by expert examiners.

Page 1. Reduce the background of the abstract to only one/two sentences and include in the methods of the abstract the name of the nine devices.

Page 3. What does mean “CE label” European community label?

Page 3. In the text of the 2.2 HHUS Devices, it is mentioned: “For three devices, no dealer contact could be established (Table 1, columns 5, 7, and 11)”. Is not row  5, 7, and 11? This is because the table 1 has only 6 columns and  14 rows. Maybe it is better to include the name of the three devices instead of referring the rows of the table.

Page 3. In the text of the 2.2 HHUS Devices, it is mentioned: “and the remaining two manufacturers (Table 1, columns 1 and 6)…”. Is not row  1 and 6? This is because the table 1 has only 6 columns and  14 rows. Maybe it is better to include the name of the two devices instead of referring to the rows of the table.

Page 3. The Table 1 title should be: “ HHUS devices requested” or similar. The rest of the text should be moved to the bottom of the table as a legend

Page 3. Table 1 refers “…the devices marker with an asterisk (*) were included in the study due to lack of response from the manufacturer or because no devices were available at the time of the study” I think that the text should be changed to “ were NOT included in the study…

Page 6. Table 2 uses average and standard deviation. Confirm if there is a normal distribution of the data. If this is not the case, use median and IQR

Page 10: Figure 6. Confirm if the letters correspond with the name of the device, a:  Vscan Air, b: Chison sono eye… If this is not the case, use the legend to name the letters.

Page 11: About section 4.2 How many examiners change the presets of the automatic mode of the devices during the examination? Does everybody change the parameters of the devices?

Page 12: About section 4.2. How much time of examination was employed? Was there any problem with the warming of the devices because of the time?

Page 13: Page 13. Rewrite the conclusion of the paper. The conclusion of the abstract is more adequate to the paper than the conclusion on page 13.

Author Response

(The authors gave the same response as above.)

Reviewer 3 Report

Comments and Suggestions for Authors

Manuscript ID: diagnostics-3158066

Title: Prospective Comparison of Nine Different Handheld Ultrasound (HHUS)

Devices by Ultrasound Experts with Regard to B-Scan Quality, Device Handling

and Software in Abdominal Sonography

Strengths: It’s an interesting topic. A prospective study by 12 examiners for the assessment of 9 ultrasound devices, with rating analysis of several variables.

Weaknesses: The study design is not adequate.

Specific concerns:

Of “Both devices differed significantly (p > 0.01)” in Abstract. It’s wrong that p > 0.01.

The study design of such study should include assessment of intra/inter-raters agreement, and intraclass correlation coefficient(ICC), otherwise, the methods are not sound and not rigorous, the results are not reliable, and the conclusions are not sound and reliable.

There were 12 examiners joining in this study, and the variables ranked last and first one in each group of the data obtained from different analyses should be ruled out, namely that data of 10 of 12 examiners were enrolled for last analysis, with the aim to reduce bias.

Of “Categorical variables were compared using Fisher’s exact test” in the Statistical Analysis. It’s not accurate, for Fisher’s exact test is not suitable for the analyses of all categorical variables.

The reviewer wondered: (Table 2) The differences of the Weight of the transducer, Depth of acoustic penetration, Gain, and so on among different ultrasound devices should not be significant in all.

Author Response

(The authors gave the same response as above.)

Round 2

Reviewer 3 Report

Comments and Suggestions for Authors

Strengths: It’s an interesting topic. A prospective study by 12 examiners for the assessment of 9 ultrasound devices, with rating analysis of several variables. The reviewer has read the revision, and appreciate the working.

Weaknesses: The statistical analysis was not adequate.

Specific concerns:

The study design of such study should include assessment of intra/inter-raters agreement, otherwise, the methods are not sound and not rigorous, the results are not reliable, and the conclusions are not sound and reliable.

There were 12 examiners joining in this study, and the variables ranked last and first one in each group of the data obtained from different analyses should be ruled out, namely that data of 10 of 12 examiners were enrolled for last analysis, with the aim to reduce bias.

Based on page 5: The assessment was based on a 5-point Likert scale (5 = very good; 4 = good; 3 = satisfactory; 2 = sufficient; 1 = unsatisfactory), the variables in Table 2 should be expressed as ordinal categorical variables,using median(interquartile range), other than using Mean ± SD.

The reviewer wondered: (Table 2) The differences of the Weight of the transducer, Depth of acoustic penetration, Gain, and so on among different ultrasound devices should not be significant in all.

The reviewer suggested a veteran statistician to verify and check the statistical analyses. Only the statistical analyses are adequate and right, can the results be correct and rigorous, and can the conclusion be rational and convincing.

The introduction, results and discussion need to be revised and optimized after the statistical analyses.

Some references were not necessary, and the number of references had better be decreased.

Author Response

Dear reviewer, thank you again for re-reviewing our manuscript. Please find attached a P2P R2 of your comments, which we hope we have addressed and implemented to your satisfaction. 

Round 3

Reviewer 3 Report

Comments and Suggestions for Authors

The references included were a little more than good, and the quotations can be optimized. Please check the consistency between the "Abstract" and the "Text body".

Author Response

Thank you very much for your advice to double-check the consistency between the abstract and the text-body. In terms of the introduction, methods, and conclusions, we saw very good agreement and refer to the adjustments made previously - in line with previous reviewer recommendations. In the results, there was a minor discrepancy in the naming of the best devices. For these reasons, we adjusted the ranking of the best devices in the overall evaluation once again (exchanging “SonoEye Chison” and “Vscan Air), so that there is now complete consistency between the abstract, results table, and text body. We have also removed further references to comply with your recommendation.